# Improved Methods for Predicting the Financial Vulnerability of Nonprofit Organizations

**Gila Burde** [ID]

Department of Management, Ben-Gurion University of the Negev, Beer-Sheva 8410501, Israel;
ladab@post.bgu.ac.il

**Abstract:** Using hazard analysis procedures, this study undertakes a longitudinal examination of Israeli Nonprofit Organizations' (NPOs') financial vulnerability arising from governmental funding instability. Funding instability is characterized by time-at-risk, which measures the level of financial instability faced by an NPO and reflects the different funding situations it encounters. The vulnerability is expressed by the hazard rate (HR), which measures the speed at which NPOs' close at a given point in time. The probability of an NPO failure is then estimated. The improvements presented in the current work are concerned with the methods of estimation of time at risk, which is a key variable in the hazard analysis, and testing a robustness of the method. The generalized time-at-risk, which measures the "level of instability" more consistently reflecting different situations encountered by a NPO, is introduced. The definition of generalized time-at-risk contains arbitrary coefficients whose values the current study determines using some optimization procedure. The optimization incorporates the idea of testing a possibility of using the results for predicting financial vulnerability by dividing the set of 2660 NPOs into two approximately equivalent samples. The coefficients in the time-at-risk definition are optimized by minimizing the average distance between the HR–time-at-risk curves based on these two samples.

**Keywords:** hazard analysis; financial failure; time-at-risk; not-for-profit organizations

## 1. Introduction

The prediction of financial vulnerability and bankruptcy is of major economic importance. Financial vulnerability is an organization's susceptibility to financial problems. Whether or not an organization is susceptible to financial problems is of concern to all stakeholders of the organization, because financial problems might not allow an organization to continue to meet its objectives and provide services.

Many studies have focused on how to improve the accuracy of failure models. Beaver (1966) was the first to employ financial ratios to predict financial failure in distinguishing failed firms from non-failed ones using comparisons of means of various financial ratios. Altman (1968) followed with the Z-Score, based on predictors with the highest predictive power in a Multivariate Discriminant Analysis model where the probability of bankruptcy increases as the Z-Score decreases. Ohlson (1980), and Zmijewski (1984) used multinomial choice techniques, including Probit and Maximum Likelihood Logit. Ohlson's (Ohlson 1980) one-year prediction model is based on an O-Score that uses coefficients as proxies for financial distress. These studies were undertaken in the context of the for-profit sector. Tuckman and Chang (1991) developed a theory specifically designed to assess NPOs' financial vulnerability. Accordingly, an NPO should be considered financially vulnerable if it is liable to curtail its services instantaneously when it experiences a financial shock such as the loss of a major donor or an economic downturn (Tuckman and Chang 1991). Greenlee and Trussel (2000) developed a model to predict NPOs' financial distress by applying for-profits prediction methodologies. Hager (2001),

Trussel (2002) and Trussel and Greenlee (2004) followed by employing accounting ratios to estimate NPOs' financial distress. The above models for predicting financial distress and bankruptcy are mostly predicated on single-period or cross-sectional data (Duffie et al. 2007). As Greenlee and Trussel (2000) state, these models generally use financial data as their financial vulnerability predictor variables, financial information pertaining to at least one year prior to the onset of financial vulnerability. The Hazard model (Cox 1972) has traditionally been applied in the field of medical research where duration until death or duration until appearance or reappearance of a disease is usually the event of interest. The growing popularity of the use of hazard models to predict corporate failure has motivated us to undertake this study. Since the seminal work of Shumway (2001), the use of the hazard rate modelling technique has become a popular methodology in bankruptcy prediction studies (see among others (Chava and Jarrow 2004; Campbell et al. 2008; Gupta et al. 2015, 2017)). According to Shumway's (Shumway 2001) the hazard bankruptcy model involves a survival analysis (Balcaen and Ooghe 2006) rather than a cross-sectional design. By ignoring the fact that firms change overtime, cross-sectional models produce biased bankruptcy probabilities (Bauer and Agarwal 2014) and inconsistent estimates of the probabilities that they approximate (Shumway 2001). The superiority of hazard models in predicting binary outcomes is well documented in the literature (see among others (Beck et al. 1998; Shumway 2001; Allison 2014)). The effectiveness of hazard models as applied to the corporate sector, that has been demonstrated in the above discussed studies, and universality of the hazard rate modelling technique suggests applying that technique to the non-profit sector.

The purpose of the current study is to develop the improved methodology of using hazard analysis in examination of financial vulnerability. Based on the procedure developed in Burde (2012) and Burde et al. (2016) the study undertakes a longitudinal examination of Israeli Nonprofit Organizations' (NPOs') financial vulnerability arising from governmental funding instability. Funding instability is characterized by time-at-risk, which measures the level of financial instability faced by an NPO and reflects different funding situations it encounters, and the vulnerability is expressed by the hazard rate (HR), which measures the speed at which NPOs' close at a given point in time. The probability of an NPO failure is then estimated.

The research was based on data collected by the Israel Central Bureau of Statistics on Israeli NPOs that had obtained central governmental funding through a ministerial support grant at least once during an eleven-year period (1997–2007), inclusive (data were provided courtesy of the Israeli Centre for Third Sector Research, the ICTR). ICTR database was amended to extract a sample meeting the needs of the present study. First, only NPOs that had obtained governmental funding through a ministerial support grant at least once during an eleven-year period (1997–2007) inclusive were selected (6216 NPOs). Next, NPOs that did not undergo funding instability were excluded. It is also evident that NPOs closed before 1997 should be excluded from the sample (meaning, NPO failures in the sample could begin only from 1998). Initially, we aimed at both privately funded and governmentally funded NPOs. Since data about non-governmental funding are unavailable and when they rarely are it is impossible to account for consecutively registered longitudinal data. Therefore we resorted to governmentally funded NPOs, for which we could obtain sequential (yearly) data. The sample, obtained as the result of such a selection, consists of 2660 NPOs.

Reasoning behind using the sample of exclusive NPOs receiving a grant from the government, is that the uncertainty related to funding activities is impacted significantly by the government. Public funding in Israel constitutes more than half of the NPOs' total income (CBS, Statistical Abstract of Israel). These support grants are transferred to NPOs expecting them to "further the policy" of granting ministries. The large sums transferred illustrate the high dependence of the third sector on public support. Hager et al. (2004) found that the NPOs dependent on government funding are more vulnerable than those existing at the expense of other funding sources. This supports our focus on studying vulnerability of NPOs with government funding.

Public funding of Israeli NPOs takes two major forms: contracts and ministerial support grants. Only one funding source is considered, ministerial support grants, as these are the major

central-government funding mechanism of relevance to Israeli NPOs. Though these supports are not necessarily the main source of income, the reason underling the choice of solely one form of governmental support is that still they constitute an essential income source since they are flexible in terms of being used for various purposes including current expenses. Additionally, for many NPO's they constitute the main source of income. This support however, rarely comes without strings, one of which is the continuation of financial support following political changeover. Steadiness of governmental support therefore determines to a great extent NPO's resource munificence (Yeager et al. 2014) and influences organizational mortality (Hager et al. 2004).

The improvements presented in the current work concern with the methods of estimation of time-at-risk, which is a key variable in the hazard analysis, and testing the possibility of exploiting the results for predicting financial vulnerability. In the methodology developed, new formal definitions for time-at-risk with the intent of incorporating different funding situations are applied. For example, the NPO fails to obtain grants only once during the period of interest, or, after an unfunded period, the NPO successfully obtains grants throughout the rest of the period, or the NPO fails to obtain grants numerous times during the period of interest, and so on. As distinct from the common time-at-risk variable, which represents the length of time elapsed from the first break in obtaining grants until organisational failure or the last available data point, the new generalized time-at-risk variables do not represent a specific time period and therefore are measured in 'conditional years'.

The possibility of using the research results for predicting financial vulnerability was tested by dividing the set of 2660 NPOs into two sub-samples that are approximately equivalent in terms of their size, and in terms of the NPOs' fields of activity, function, and age distributions. The HR–time-at-risk curves based on these two samples are compared. The curve based on one of the samples is considered as a "standard" curve which may be used for prediction purposes. The curve based on the second sample then is considered as that based on what happened in reality (later data). If a prediction ('standard' curve from the first sample calculations) is close to what happened in reality (the curve from the second sample calculations) then the prediction is robust. The closer these curves are to one another, prediction is more accurate. Since the definitions of the generalized time-at-risk contain arbitrary coefficients, which should be specified before running the hazard analysis procedure, there is a possibility of adjusting the coefficients in order to make the curves closer. Of course, if curves practically merged, prediction would be most accurate but, in practise, such an extreme case has never been found in the results of the present calculations. The optimization procedure has been developed and applied in order to specify the values of the coefficients such that the average distance between the HR–time-at-risk curves based on these two samples were minimal. Then any of the curves, or the curve based on the whole sample 2600 NPOs (which always lies between the curves for subsamples), can be used for robust predictions.

## 2. Methodology

### 2.1. Outline of the Hazard Analysis Procedure

The objective of the hazard analysis procedure is to quantify the instantaneous risk that an organisation will close at time $t$. Since time is continuous, the probability that closure will occur at exactly $t$ is zero. However, there exists an observable probability that the event will occur in the interval (a full year in the present case) between $t$ and $t + \Delta t$. The probability is conditional on the firm surviving to $t$, since firms that have closed are no longer at risk of failure. The hazard function captures this relationship via the hazard rate (HR, also known as the hazard function). The HR, $h(t)$, expresses the probability that an organisation will fail within a specific time period as follows:

$$h(t) = \lim_{\Delta t \to 0} \frac{\Pr(t < T < \Delta t | T > t)}{\Delta t} \tag{1}$$

where $T$ is a nonnegative random variable denoting the time to organisational failure.

The Cox proportional hazards model (Cox 1972) asserts that the HR for the *i*th subject in the data is:

$$h_i(t|x_i) = h_0(t) \exp(x_i \beta_x) \tag{2}$$

where the regression coefficients, $\beta_x$, are to be estimated from the data. The baseline hazard function $h_0(t)$ is:

$$h_i(t|x_i) = 0 \tag{3}$$

From (2), $h_0(t)$ corresponds to the overall hazard when $x_i = 0$.

The term 'baseline HR' refers to the hazard function when all covariates are equal to zero (Klein and Moeschberger 2003). In our research, we estimated solely the baseline HR. Although the Cox model produces no direct estimate of the baseline HR, one may obtain estimates of the baseline survivor function corresponding to a baseline HR, the baseline cumulative hazard function, and the baseline hazard contributions, which may then be smoothed to estimate baseline HR itself. We used a Gaussian (normal) kernel function,

$$K(u) = (2\pi)^{-1/2} e^{-u^2/2} \tag{4}$$

where $u = \left(\frac{t - t_i}{b}\right)$ for each observed failure time, $t_i$ (Wang 2005).

## 2.2. Methods for Time-At-Risk Estimation

Time-at-risk is a key variable in hazard analysis. Various ways to estimate time-at-risk are addressed in the extant literature (Gepp and Kumar 2008, Gepp and Kumar 2015, Gupta et al. 2017, Hager et al. 2004), but they do not reflect the variety of situations encountered by organizations. In the present study, the concept of generalized time-at-risk (in units of conditional years), which measures the level of funding instability by taking into account not only duration and timing of funding instability but also some other related factors. Several possible definitions of generalised time-at-risk using that conceptualization can be introduced (see Appendix A).

## 2.3. Using the Results for Predicting Financial Vulnerability

As described in the Introduction, this possibility was tested by dividing the sample of 2660 NPOs into two approximately equivalent samples in terms of size; field of activity; function; and age distributions. Predicated on the data for one of the samples, a standard curve was formed with the view of predicting NPOs' financial vulnerability. The second sample was employed for testing a robustness of the prediction. Whenever calculations on the second sample provided results close to those predicted using the 'standard' curve obtained on the first sample, the curve may be considered robust for prediction purposes.

## 2.4. Testing the Robustness and Optimization Procedure

It is found in Burde (2012) and Burde et al. (2016) that the relationships between HR and time-at-risk are not monotonic and have an inverted U-shape curve, i.e., the hazard rate first increased with time-at-risk, reached a maximum at some value of time-at-risk and then descended (see Figure 1). These results imply that, whenever an NPO is faced with a funding instability, there is some 'critical' period (in terms of time-at-risk) when a probability of the NPO closure is maximal—after this period NPOs' financial vulnerability decreases. A possibility of using the hazard analysis results for predicting financial vulnerability is tested by dividing the set of 2660 NPOs into two approximately equivalent samples (here labelled samples 1 and 2). Then the HR–time-at-risk curves based on these two samples are calculated (an example of two HR versus time-at-risk curves is shown in Figure 1). The relative positions of the curves on the graph depend on the definition of time-at-risk and, for a specific definition, on the values of the coefficients contained in the definition. The values of the coefficients

are optimized by minimizing the average distance between the HR–time-at-risk curves based on these two samples.

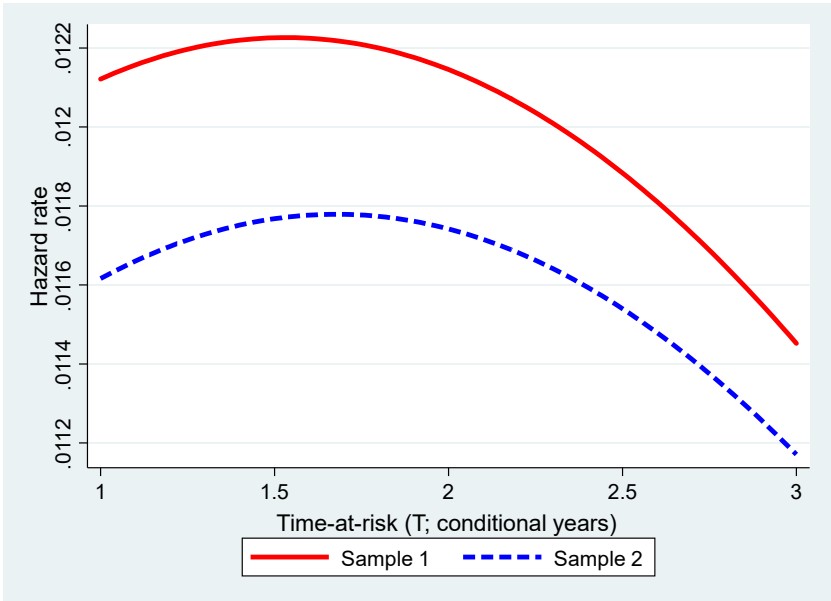

**Figure 1.** Plots of the hazard rate estimate versus time-at-risk for two sub-samples.

The optimization procedure has been applied using one of possible definitions of generalized time-at-risk (Burde 2012) T outlined in the Appendix A, namely

$$T = k_1 * N_0 + k_2 * N_1 \tag{5}$$

Here $N_0$ is the length of time that elapses from the first break in obtaining grants until the NPO fails or until the end of the period of interest whichever comes first while $N_1$ represents the number of no-grant periods of one or more years within that time period.

If $H1$ and $H2$ denote the two sets of values of the HR calculated using sample1 and sample 2 then the relative average deviation between the two sets of HR values (relative average distance between the HR–time-at-risk curves) is calculated, as follows:

$$Dev = \frac{\left[\frac{S}{101}\right]^{0.5}}{V} \tag{6}$$

where

$$S = \sum_{i=1}^{101} (H1_i - H2_i)^2 \tag{7}$$

And

$$V = \frac{\sum_{i=1}^{101} H1_i}{101} \tag{8}$$

It should be clarified that 101 is not an amount of data (the number of NPOs) in each sample (which was approximately 1300), but rather the number of points obtained from the data by some interpolation procedure in the hazard analysis package.

### 3. Example of Results

In the optimization procedure, the relative deviation Dev was calculated by keeping the coefficient $k_2$ constant while varying the value of $k_1$. The calculations were repeated for several different values of $k_2$. The results are presented in Figure 2.

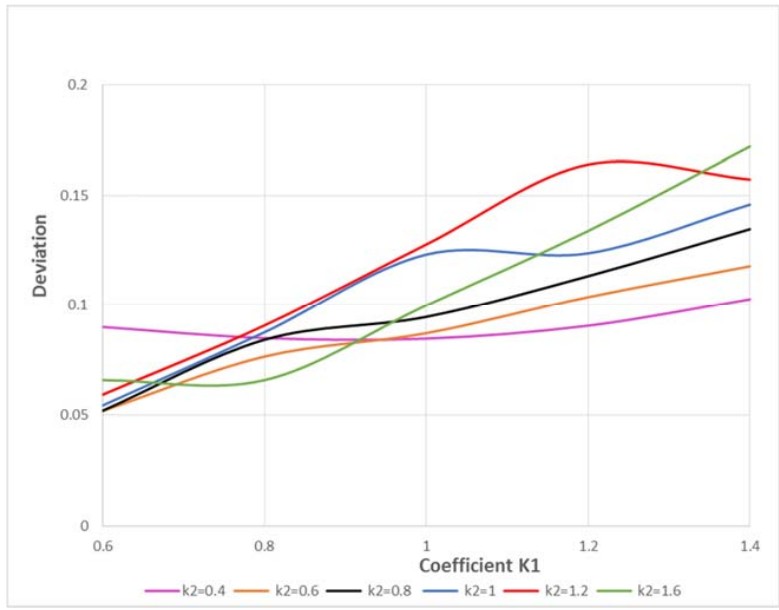

**Figure 2.** Deviation for the different constant values of the coefficient $k_2$.

It is seen that changing the values of the coefficients may both reduce and increase the deviation significantly so that, in any practical application, using the procedure allows one to choose the values of the time-at-risk parameters $k_1$ and $k_2$ in some optimal way.

### 4. Conclusions

The present study was aimed at developing improvements to the methods based on the hazard analysis that are used for predicting financial vulnerability of nonprofit organizations. The improvements are based on two new ideas: (1) introducing the generalized time-at-risk, which measures the "level of instability" more consistently than the commonly used one, and (2) dividing the sample of data into two approximately equivalent samples, while comparing the results obtained using each sample, which allows to adjust the results for the prediction purposes by optimizing the method parameters such that the difference between the results were minimal. Whenever calculations on the second sample provide results close to those predicted using the 'standard' curve obtained on the first sample, the curve may be considered robust for prediction purposes.

The results imply that incorporating those ideas into the hazard analysis procedure might significantly improve methods for predicting financial vulnerability of nonprofit organizations.

The new ideas, which have allowed us to improve the hazard analysis method as applied to the non-profit sector, are of general theoretical and methodological value and contribute to the theory and literature on the hazard method applications, in general—both in economic studies and in other areas of science (for example, medicine) where the hazard modelling technique is applied.

**Acknowledgments:** The author is grateful to the anonymous reviewers for useful comments that significantly improved the presentation.

**Conflicts of Interest:** The author declares no conflict of interest.

## Appendix A. Definitions of Generalized Time-At-Risk

In this Appendix, several possible definitions of the generalized time-at-risk, $T_0$, $T_1$, $T_2$, and $T_3$ Burde (2012) are considered. $T_0$ is the simplest way to define the time-at-risk, and does so by equating it with the length of time that elapses from the first break in obtaining grants until the NPO fails or until the end of the period of interest ($N_0$), whichever comes first i.e., $T_0 = N_0$. This definition does not reflect many situations faced by NPOs. For instance, it fails to differentiate between a situation in which an NPO fails to obtain grants only once versus numerous times during the studied period. Additionally, several consecutive no-grant years are potentially riskier for an NPO than a single year without a grant if the NPO successfully obtains grants throughout the rest of the relevant period. Consequently, the second definition for time-at-risk, e.g., $T_1$ is introduced by

$$T_1 = k_1 * N_0 + k_2 * Y \qquad (ApA\_1)$$

where $N_0$ is as defined above, whilst $Y$ is defined as follows:

$$Y = N_1 + k_3 N_2 + k_4 N_3 + \ldots . \qquad (ApA\_2)$$

where $N_1$ represents the number of no-grant periods of one or more years, $N_2$ the number of no-grant periods of two or more years, and so on. The coefficients $k_1$ and $k_2$ of $N_0$ and $Y$ in Equation (10), as well as additional coefficients $k_3$, $k_4$ and so on within the definition of Y, serve as weighting factors that consider the cumulative effect of consecutive years of failure with respect to government funding. To reduce the number of parameters that take part in the optimization procedure the coefficients $k_3$, $k_4$ and so on can be parameterized using one parameter $k_0$, as follows

$$Y = N_1 + (1 + k_0)N_2 + (1 + 2k_0)N_3 + \ldots . \qquad (ApA\_3)$$

Such a parameterization is intended to account for the fact that consecutive no-funding years should receive added weight compared to non-consecutive years.

The next formulation for time-at-risk ($T_2$) was developed to account for a scenario in which an NPO experiences periods without grants that alternate with grant-funded years. In this scenario, the number of years from the first break in grants to the end of the period (i.e., $N_0$) is of lesser importance, and thus $N_0$ can be replaced with $Y$ to yield the following formula for $T_2$

$$T_2 = Y \qquad (ApA\_4)$$

Finally, the definition of time-at-risk $T_3$ that comes closest to the notion of state funding instability, could be

$$T_3 = X \qquad (ApA\_5)$$

where $X$ is the total number of no grant years within the period of interest.

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
