# Peer review of "Improved Methods for Predicting the Financial Vulnerability of Nonprofit Organizations"

_admsci, doi:10.3390/admsci8010003_

Round 1
Reviewer 1 Report
The author has presented an interesting idea in terms of identifying the financial uncertainty of non-profit organizations. However, more justification and explanation on the applicability of models have been used in the corporate sector into the non-profit sector. In addition, the study should demonstrate a significant contribution to literature, theory, and practice.
Furthermore, the sample used for this study is exclusive NPOs receiving a grant from the government, the uncertainty related to funding activities seems to be impacted significantly by the government.
The author should also discuss further the importance and the purpose of dividing of sample into two equivalent samples.
The author should also discuss further the importance and the purpose of dividing of sample into two equivalent samples.
Author Response
Response to the comments of the first reviewer.
A. However, more justification and explanation on the applicability of models have been used in the corporate sector into the non-profit sector.
Thanks for this comment. See Introduction second paragraph, page 1 lines 34 - 67
B. In addition, the study should demonstrate a significant contribution to literature, theory, and practice.
Thanks for this comment. Comments on this issue have been added, please see Conclusions section, page 8, paragraph 2, lines 236-241
C. Furthermore, the sample used for this study is exclusive NPOs receiving a grant from the government, the uncertainty related to funding activities seems to be impacted significantly by the government.
Thanks for this comment. Discussion regarding the reasons underling the choice of exclusive NPOs receiving a grant from the government has been added. Please see Introduction, page 3, paragraph 1, lines 88-95
C. The author should also discuss further the importance and the purpose of dividing of sample into two equivalent samples.
We appreciate this remark. This issue is now discussed in more details in the Introduction, in the Methodology section, in the subsection: ‘Using the findings to predict financial vulnerability’ (page 5, par. 2, lines 173-180), and in the Conclusions section (page 8, par.1, line 233)
The author is grateful to the Reviewer for useful comments.
Reviewer 2 Report
Administrative Sciences Review for the article, “Improved methods for predicting financial vulnerability of nonprofit organizations The article discusses the use of hazard models to predict financial vulnerability in a sample of Israeli nonprofits. The models seems a reasonable mechanism for answering the research question. tionquestion is an The question is an important one for scholars but more importantly for nonprofit leaders and their boards. The importance of grant funding within the hazard model needs clarification. If grants funds are theoretically important, this aspect of the research bears more discussion, and if not, why is the data sorted on this criterion? The Hager et al paper finds that government grants influence organizational mortality, is that one of the reasons why grants are included in the model? What do you mean by privately owned NGOs? What is the ICTR database? Do you have any information regarding its validity and reliability? The nonprofit literature referenced on page 1 ln 38 (Tuckman and Chang, Greenlee and Trussel, and Hager) do discuss organizational mortality and financial vulnerability factors but are not necessarily equivalent with bankruptcy forecasting. The larger issue is that there are some nuances within the literature regarding financial vulnerability, fiscal health, organizational mortality that are masked by such a general lumping of all these studies together. Overall the discussion of relevant literature and theory seems slim. The Hager et al 2004 article references on page 3 ln 199 discusses a hazard model but the authors say there are various ways to reference time-at-risk but only reference one article. There should be other references here. Figure 1 seems to have part of header for the x-axis missing. Appendix D: Are these definitions the authors ideas or are they coming from other literature? My guess is the latter in which case the other literature needs to be cited within the body of the paper and in the appendix. Proofreading errors occur here and there. Some examples include: • Time-at-risk appears both with and without hyphens • Pg 2 ln 64 “sake analyzing”, ln 77 “to test a robustness of the method” • References: NonProfit Management & Leadership should be Nonprofit I wish the authors the best as they proceed with this research!Author Response
Response to the comments of the second reviewer
A. The importance of grant funding within the hazard model needs clarification. If grants funds are theoretically important, this aspect of the research bears more discussion, and if not, why is the data sorted on this criterion? The Hager et al paper finds that government grants influence organizational mortality, is that one of the reasons why grants are included in the model?
Discussion regarding the reasons underling the choice of solely one form of governmental support has been added. Please see page 3, paragraph 2, line 97 starting with “Only one funding source is considered,…”).
B. What do you mean by privately owned NGOs?
Thanks for this comment, “privately owned” has been corrected to “privately funded”. Please see page 2, line 83 –par. 3, 6th line from the end of the page.
C. What is the ICTR database? Do you have any information regarding its validity and reliability?
Information regarding the ICTR database has been added, please see page 2 line 75 par. 3 starting with “The research was based on data collected by the Israel Central Bureau of Statistics…”
D. The nonprofit literature referenced on page 1 ln 38 (Tuckman and Chang, Greenlee and Trussel, and Hager) do discuss organizational mortality and financial vulnerability factors but are not necessarily equivalent with bankruptcy forecasting. The larger issue is that there are some nuances within the literature regarding financial vulnerability, fiscal health, organizational mortality that are masked by such a general lumping of all these studies together. Overall the discussion of relevant literature and theory seems slim.
The discussion of the relevant literature is extended. Please see page 1 par. 2
E. The Hager et al 2004 article references on page 3 ln 199 discusses a hazard model but the authors say there are various ways to reference time-at-risk but only reference one article. There should be other references here.
Thanks for this comment. Other references have been added (Gepp and Kumar 2008, Gepp and Kumar 2015, Gupta, Gregoriou and Ebrahimi 2017..). Please see page 4 last paragraph, line 165(Line 3 from the end of the paragraph).
The phrase “Hager et al. (2004) estimated time-at-risk as the number of years after exiting from the panel” has been deleted
F. Figure 1 seems to have part of header for the x-axis missing.
Header for the x-axis has been corrected, “T; conditional years” instead of “T; years”. Please see Figure 1 page 6
Please see also Figure 2, page 7. Header for the x-axis has been added: “Coefficient k1”
G. Appendix A: Are these definitions the authors ideas or are they coming from other literature? My guess is the latter in which case the other literature needs to be cited within the body of the paper and in the appendix.
Many thanks for this comment. These definitions are the author’s ideas. The concept of generalized time-at-risk (in units of conditional years), which measures the “level of funding instability” was introduced in Burde, G. (2012). See Appendix A.
H. Proofreading errors occur here and there. Some examples include:
• Time-at-risk appears both with and without hyphens
• Pg 2 ln 64 “sake analyzing”, ln 77 “to test a robustness of the method”
• References: NonProfit Management & Leadership should be Nonprofit
Proofreading errors have been corrected.
The author is grateful to the Reviewer for such a careful reading and useful comments.